# Defining Religion: Durkheim and Weber Compared

**Robert Launay**

Department of Anthropology, Northwestern University, Evanston, IL 60208, USA; rgl201@northwestern.edu

**Abstract:** Emile Durkheim began The Elementary Forms of the Religious Life with an injunction: "In order to identify the simplest and most primitive religion that observation can make known to us, we must first define what is properly understood as a religion". Almost simultaneously, Max Weber would begin the long section on the sociology of religion in his unfinished work Economy and Society by insisting, "To define 'religion', to say what it *is*, is not possible at the start of a presentation such as this. Definition can be attempted, if at all, only at the conclusion of the study" (1978, p. 399). Durkheim's insistence and Weber's reticence are equally surprising. By and large, Durkheim's writings are relatively sparing of definitions. He did not generally bother to define words that were already in common currency. "Religion" is unquestionably the most notable counterexample. On the other hand, Weber was far more scrupulous—one might even say obsessive—about defining terms that were not specifically his own, including "capitalism", "class", and "bureaucracy" to select only a few examples. Durkheim's long disquisition on the definition of "religion" was as radically atypical of his modus operandi as was Weber's avoidance. The question of religion's definition provides a fruitful window into their opposing analyses of non-European societies as a means of characterizing European modernity, ways that derive in important respects from early modern depictions of "savages" and "Orientals".

**Keywords:** Durkheim; Weber; theories of religion; sociology; anthropology; non-European religions; totemism



Emile Durkheim published The Elementary Forms of Religious Life (Durkheim 1995) in 1912, shortly before the outbreak of the First World War. Almost simultaneously, Max Weber wrote the book-length section on the sociology of religion for his massive treatise Economy and Society (Weber 1978), though it was only published posthumously. Clearly, "religion" was central to the preoccupations of the two most prominent sociologists of the early twentieth century. In many respects, these works demonstrate antithetical approaches. Indeed, neither author makes any reference whatsoever to the other in these two volumes. Behind these differences lie important convergences. Both authors take the concept of "religion" and, equally important, its universality for granted. This is entirely unsurprising; the awareness that "religion" is a European concept with its origins in the Early Modern era dates only to the late twentieth century. The problem is that non-European "religious" ideas and practices are absolutely central to both books and to the theoretical preoccupations of both authors. Each author, though in a very different way, uses non-European examples as a foil to highlight what he sees as the defining features, not so much of European religion but more broadly of European modernity per se. In the process, both authors attempt, though with only limited success, to break away from nineteenth-century evolutionary models of progress that legitimate European hegemony over the rest of the world. The break with evolutionary approaches is most starkly exemplified by each author's insistence of the rationality of non-European religious ideas, their refusal to condemn non-Europeans as irrational, even as they refuse to acknowledge them, to use Fabian's (1983) term, as "coevals". In the process, both Durkheim and Weber draw on different Enlightenment paradigms for characterizing non-Europeans: polemical depictions of "savages" for Durkheim and "Orientals" for Weber.[1]

Emile Durkheim began The Elementary Forms of Religious Life with an injunction: "In order to identify the simplest and most primitive religion that observation can make known to us, we must first define what is properly understood as a religion" (Durkheim 1995, p. 21). Almost simultaneously, Max Weber would begin the long section on the sociology of religion in his unfinished work Economy and Society by insisting, "To define 'religion', to say what it *is*, is not possible at the start of a presentation such as this. Definition can be attempted, if at all, only at the conclusion of the study" (Weber 1978, p. 399). Any reader who might expect a definition at the end of the section will be disappointed. Durkheim's insistence and Weber's reticence are equally surprising. By and large, Durkheim's writings are relatively sparing of definitions. Of course, he was careful to define terms that constituted his specific analytical vocabulary: "social facts", "*conscience collective*", and "anomie", for example. However, he did not generally bother to define words that were already in common currency. "Religion" is unquestionably the most notable counterexample. Of course, Weber also had his own theoretical vocabulary: "ideal types" and "charismatic authority". On the other hand, Weber was far more scrupulous—one might even say obsessive—about defining terms that were not specifically his own, including "capitalism", "class", and "bureaucracy" to select only a few examples. Durkheim's long disquisition on the definition of "religion" was as radically atypical of his modus operandi as was Weber's avoidance.

The very fact that the concept of "religion" is a specifically modern European construction (See Asad 1993; Nongbri 2013) makes the question of whether and how to define it of particular strategic importance. Talal Asad's (1993) mordant critique of Clifford Geertz's definition of religion as a system of symbols elegantly demonstrates the point. Instead, Asad insists that "a transhistorical definition of religion is not viable" (Asad 1993, p. 30). By proposing a universal definition, Geertz inappropriately applies his own unexamined ideological assumptions to the analysis of other traditions. Although Asad is particularly concerned with liberal misconstruing of the Islamic tradition, he deliberately includes examples from medieval Christianity in order to suggest that liberal approaches to "religion" even fail to understand their own religious tradition. Indeed, Horii (2019) has elaborated a similar critique of Durkheim's definition and even what he considers to be Weber's implicit definition, situating them within the colonial ideologies of the late nineteenth century German empire and the French Third Republic.

This article is not specifically concerned with the actual content of the definitions of religion. Weber's is in any case nonexistent. There is such an extensive literature on Durkheim's opposition of Sacred and Profane that I doubt I have anything new to add. Nor will I assess in detail the scholarship and accuracy of their (mis)representations of non-European societies and religions. In the century since their publication, there have been numerous, often scathing critiques, in Durkheim's case virtually since the book appeared in print. Rather, this essay examines the relationship between their decisions to define or not to define "religion", the ways of defining that they explicitly or implicitly exclude, and their very different strategies of deployment of non-European examples in order to place their understandings of the predicament of modern Europe in relief. Such strategies were intended as a break with their immediate predecessors in the late nineteenth century. Even so, they incorporate significant continuities, not only with recent thinkers but also, perhaps more profoundly, with the Enlightenment.

## 1. Religion Defined

Durkheim felt impelled to justify his decision to define religion, particularly since the question took up the entire first chapter of the book. He began, plausibly enough, by arguing that, without a definition, we risk overlooking true examples of religion, or else including phenomena that are not really religious. Instead, we must rid ourselves of any preconceptions (easier said than done) and instead "consider religions in their concrete reality and try to see what features they may have in common" (Durkheim 1995, p. 22). For someone with as sophisticated a philosophical awareness as Durkheim, this is remarkably

illogical. Having just warned us that we need a definition in order to identify genuine cases, he then instructs us to compare all cases in order to extract a definition. Had Durkheim taken such a methodology seriously, the logical contradiction would indeed be scandalous. In fact, the definition at which Durkheim eventually arrived, centering on the distinction between the categories of "sacred" and "profane", hardly seems a plausible product of inductive comparison. The assertion that such a distinction can unequivocally and fruitfully be applied to all religious traditions is, to say the least, contentious (See, for example, (Allen 2013; Evans-Pritchard 1965; Goody 1961; Stanner 1967)).

While Durkheim's methodological injunctions could hardly be taken literally, they nonetheless served an important purpose. Ridding oneself of one's preconceptions and taking all religions into account, however impossible in practice, was meant to rule out any paradigm explicitly or implicitly centered on the European Judeo-Christian religious tradition. Instead, Durkheim chose totemism as a paradigmatic religion, a move that in the intervening century has lost its novelty but which, at the time, was radically unconventional. Durkheim's decision on how to define and, even more revealingly, how not to define religion were all framed with the objective of making such a choice plausible, if not necessary. For this purpose, rejecting European religiosity as a model was merely the first step.

Even so, why would such a démarche require a definition in the first place? Durkheim provided the answer in the very first paragraph of the chapter. No less an authority than Sir James Frazer had denied that totemism was a religion. Ever since the publication of The Golden Bough in 1890 (Frazer 1890), Frazer was unquestionably one of the most prominent and influential anthropological thinkers of his day. In 1910, only three years before the publication of The Elementary Forms (Frazer 1910), Frazer had published a massive (four thick volumes!) study, Totemism and Exogamy: A Treatise on Certain Early Forms of Superstition and Society. Durkheim had actually wanted to translate an earlier and much shorter version of the work but was beaten to the punch by Arnold van Gennep (Thomassen 2016, p. 176). As the subtitle suggests, Frazer characterized totemism as a "superstition", as something that could not be taken seriously at face value. Indeed, for Frazer, religion was not a universal nor a primordial human phenomenon. In The Golden Bough, he had postulated a chronological sequence from magic to religion to science. Religion had supplanted magic once humans realized its inadequacies, just as science was replacing the inadequacies of religion. Totemism, Frazer argued, belonged to the realm of magic rather than of religion. If only to refute the claims of Frazer, whose importance at the time must not be underestimated, Durkheim needed a definition to demonstrate not only that totemism could legitimately be called a religion but that its analysis could serve to determine the characteristics of religion in general.

Durkheim went to great lengths not only to elaborate his definition of religion but also to rule out others. These arguments, too, were clearly framed with totemism in mind. He specifically rejected two definitions. The first linked religion to the notion of the supernatural. Durkheim's extremely astute and perceptive objection was that the concept of "supernatural" was recent and historically contingent. (Ironically, Durkheim did not imagine that the same could be argued for the very concept of "religion".) Specifically, the idea of the supernatural depended on "an awareness that there is a *natural order of things*, in other words, that the phenomena of the universe are internally linked according to natural relationships called laws" (Durkheim 1995, p. 25). While Durkheim never mentioned the name of Newton, it is clear that he conceived the supernatural as an integral feature of a post-Newtonian conceptual universe. The supernatural was a residual domain consisting of whatever could not be explained in scientific terms. In this sense, it was not at all the equivalent of the extraordinary, the marvelous, or even the miraculous, phenomena that might diverge from quotidian experience but hardly from the "natural order of things". In any case, Durkheim added, religion is far more concerned with quotidian experience than with the extraordinary.

However persuasive, Durkheim's objections masked his ultimate motivations. Totemism as a religion is entirely preoccupied with natural phenomena, with species—kangaroos, emus, witchetty grubs—or even with celestial objects such as the sun and the moon. The supernatural, as opposed to the natural, is conspicuously absent from totemism. Defining religion in terms of the supernatural would clearly exclude totemism from its purview.

This was equally true of the second definition excluded by Durkheim, one which categorized religion as the worship of divinities or, more generally, in language suggested by Edward Tylor, as "spiritual beings". Durkheim's rebuttal is here on shakier grounds. He relied in particular on late nineteenth century accounts of Buddhism.[2] Divinities, he suggested, were incidental to if not absent from Buddhism or for that matter, he added, to Jainism. (He could easily have included Confucianism for good measure.) He conceded that, in some but by no means all Buddhist communities, the Buddha was worshipped as a divinity. Even so, Buddhism was never reducible to, much less identifiable as, worship of the Buddha. Whether or not Durkheim's categorization is accurate, his argument involves a logical contradiction. His call for a definition of religion in the first place asserted that only a correct definition can inform us what ought to be included or excluded from the domain of religion. If a definition of religion in terms of the centrality of the worship of divinities would not include Buddhism, must we reject the definition? It would be equally logical to conclude that, in spite of our misconceptions, we must reject the proposition that Buddhism is a religion. This is not an outcome that Durkheim was prepared to accept. He preferred to reject the definition rather than Buddhism.

Once again, his objections were formulated with totemism in mind. Although the concept of totemism was first introduced in anthropology by McLennan as "the worship of animals and plants", there was never any suggestion that natural species were treated as divinities. The absence of divinities or even anything that might plausibly be labelled "spiritual beings" seemed even more definitive for totemism than for Buddhism.

Strikingly, these were the only two definitions that Durkheim took the trouble to rule out. Were no other definitions available? In fact, in the following two chapters on "The Leading Conceptions of the Elementary Religion", Durkheim criticizes different theoretical accounts of the origins of religion that implicitly provide yet other bases for definition. In the first place, he rejects the notion of "animism" as proposed by Edward Tylor and Herbert Spencer. Both theorists attributed the origins of religion to primitive explanations of dreams in terms of the existence of a double, or soul. For Tylor, such souls could equally be attributed to inanimate objects and, for Spencer, to spirits of the dead. In such schemes, religion was ultimately reduced to a system of explanation, albeit an erroneous one. Such schemes had their roots in the positivism of Auguste Comte, in the assertion that all knowledge began with theology and proceeded to metaphysics and ultimately to science. From the very outset of the book, Durkheim rejected such an approach: " . . . it is a fundamental postulate of sociology that a human institution cannot rest on error and falsehood" (Durkheim 1995, p. 2). To categorize religion as a system of explanation was to suggest that it was simultaneously analogous to and inferior to science. However, Tylor's and Spencer's theory of animism in no way ruled out "totemism" as a religion. Whatever Durkheim's theoretical objections, definitions of religion as a system of explanation did not call "totemism" into question per se.

The same could be said of "naturism", the theory expounded by Friedrich Max Müller. For Max Müller, religion emerged out of the sense of awe that nature inspired in primitive humans. Such a theory bears close resemblance to Rudolf Otto's (1923) influential definition of religion as centered on "the holy", on the individual's experience of the numinous, admittedly published a few years after Elementary Forms. Durkheim took pains to reject conceptions of religion derived from individual experience or individual consciousness. However much Durkheim objected to definitions of religion along such lines, they did not rule out in principle any inclusion of totemism. As with "animism", he reserved his critique of such theories and the definitions they implied in the following chapters.

The leap from definitions Durkheim rejected to the one he proposed comes as a startling conceptual shift. His discussions of definitions to reject revolved, implicitly but strategically, on the criteria that determined what could or could not be included in the rubric "religion". The definition Durkheim ultimately proposed hardly seems to serve a similar purpose:

> "A religion is a unified system of beliefs and practices relative to sacred things, that is to say, things set apart and forbidden—beliefs and practices which unite into one single moral community called a Church, all those who adhere to them". (Durkheim 1995, p. 44)

This is a radically inclusive definition, of little if any use for determining what should be excluded from the domain of religion. For example, is nationalism a religion? Durkheim's famous metaphor of the totem as the flag of the clan suggests that, for Durkheim in particular, the question was hardly nonsensical and on the eve of the First World War was retrospectively rather ominous. Durkheim's definition, rather than ruling out various phenomena from the purview of "religion", spelled out in no uncertain terms Durkheim's lines of analysis. It is hard to imagine anyone accepting such a definition without being, to one extent of another, a theoretical Durkheimian. One can easily accept Tylor's definition of religion as belief in spiritual beings while adamantly rejecting Tylor's theories. On the other hand, accepting the notion that religion is intrinsically about the distinction between the sacred and the profane unequivocally marks one as a follower of Durkheim.

While Durkheim's definition is of little help in determining what in particular is or is not *a* religion, there is one distinct domain where he took pains to specify what *is not* religion: magic. Like religion, magic, according to Durkheim, depends on the division of the world into sacred and profane elements. Unlike religion, it neither depends on nor creates a moral community; a magician has a clientele, not a Church. The distinction has important implications. The division of the world into sacred and profane was, for Durkheim, entirely dependent on the existence of a moral community. Seen in these terms, magic clearly derived from religion, and not vice versa, as Frazer had famously claimed. At the very end of Elementary Forms, Durkheim again sought to refute Frazer's contention that science must replace religion just as religion replaced magic. He insisted, on the contrary, that "the essential notions of scientific logic are of religious origin" (Durkheim 1995, p. 431). Rather than a transitory (if persistent) phase of human evolution, religion, Durkheim contended, is a universal—arguably the most fundamental—feature of the human experience.

Durkheim's discussion of definitions of religion thus takes us from the realm of the particular—totemism—to that of the most universal. The trajectory was entirely deliberate: Durkheim focused on the analysis of totemism in order to discover the universally human,[3] specifically taking pains to reject any definition of religion that might exclude totemism. However, no exercise in definition can explain why totemism should loom so large in Durkheim's scheme. Admittedly, he asserted from the outset that he sought out "the simplest and most primitive religion" he could identify.[4] The most obvious motive of such a quest, particularly in the late nineteenth or early twentieth century, would have been a search for the origins of religion. Durkheim flatly rejected such an approach. He was not interested in determining whether all religions, modern or otherwise, ultimately derived from a totemic prototype. Instead, his search for simplicity was methodological. The simplest religion was presumably the most transparent, the one whose underlying principles were the most discernable (at least for the sociologist). It was also radically ahistorical. Its analysis required no knowledge of its antecedents, in the sense that Christianity is not fully comprehensible without reference to Judaism. Indeed Robertson Smith (1972), whose work was a seminal influence for Durkheim, argued that Judaism (and consequently Christianity) ultimately derived from totemism.

Needless to say, Durkheim's characterization of totemism as a "simple" religion is radically problematic. Even before the publication of Elementary Forms, Goldenweiser (1910) had published a long and important article demonstrating that the different features that presumably characterized "totemism" varied independently from one another and

were rarely found together. He stopped short, but only barely, of concluding that totemism was useless as an analytical category. Indeed, Goldenweiser (1917) published a devastating critique of Elementary Forms, as did Arnold van Gennep (2017).[5] Van Gennep criticized Durkheim's uncritical use of the Australian ethnography and his highly selective use of example to support his prior theoretical claims. Goldenweiser came to similar conclusions about his use of North-American ethnographies. In any case, subsequent ethnographers have hardly found the Australian religions that constitute the core of Durkheim's analysis to be in any way transparent or simple. The notion that Australian (or any other) religion exists in some sense outside of history is a product of our prejudices and our ignorance, with no basis in reality. Indeed, Franz Boas (1916) insisted on the historicity of "totemism" as a phenomenon in different societies, making direct reference to Durkheim's theories.

Be this as it may, the way in which Durkheim deployed the concept of totemism was radically original. Late-nineteenth-century theorists—Spencer, Tylor, and Frazer most notably—all stressed the relative irrationality of "primitive" religions such as totemism. Their primary concern was to highlight the difference between such religions (if not all religion in general) and secular modernity embodied by science. Durkheim's use of totemism to embody the universal features of religion stressed on the contrary the resemblances between primitive religion and modern thought.[6] Native Australians were like us, though in their "primitive" simplicity, not *just* like us. Paradoxically, for Durkheim, the rationality of totemism was lost on the Australians, apparent as it might be to the sociologist. Its rationality existed at the social but not the individual level. As van Gennep (Thomassen 2016), as well as American ethnographers under the aegis of Franz Boas, realized, indigenous peoples were fully capable of agency, reflection, and creativity. Still, Durkheim was also reacting against the presuppositions of evolutionary anthropology, carefully crafting his definition to encompass Australians as well as modern Europeans, both of whom could be excluded by theorists such as Frazer at opposite ends of a developmental continuum, as pre- or post-religious.

## 2. Religion Undefined

If Durkheim elaborated in detail how to define religion, Weber obviously had nothing to write about how he did not define it and little about why. He dismissed the enterprise succinctly: "the essence of religion is not even our concern (Weber 1978, p. 399)". Arnal and McCutcheon's (2013, p. 17) suggestion that, for Weber "the 'stuff' of religion is so obvious and self-evident that it can be identified and studied without even knowing ... 'what it is' "(cited in Horii 2019, p. 28) is unconvincing. If the category were so self-evident, why would he have opened his study with an admission that he was not defining it, much less dangling the (unfulfilled) possibility that he might define it at the end of the work? Such a dismissal might be unremarkable in another thinker, but the notion of "ideal type"— closely related if not identical to the notion of "essence"—was absolutely central to Weber's sociology.[7] Weber introduced the concept to distinguish between the aims and methods of sociological as opposed to historical analysis. The aim of history, Weber argued, was to provide explanations of unique events. Sociology, on the other hand, sought to uncover "generalized uniformities of empirical process" (Weber 1978, p. 19). Such generalizations were made possible in terms of ideal types, that is to say, abstract constructs that were only partially and imperfectly realized in specific empirical instances. It is hard to see how the abstract meaning of a concept is different from its "essence" and consequently why "religion" should not be an ideal type and, as such, subject to the kind of precise definition that Weber considered so important. Indeed, Clifford Geertz (1973, pp. 93–135), who prominently acknowledged the extent of his theoretical debt to Weber, formulated and elaborated just such a definition of "religion".

However, ideal types, in Weber's sociology, cannot be reduced to the way in which he defines or describes concepts but rather in the way he deploys them. In Weber's writings, ideal types are never found in isolation but rather in contrasting pairs or more occasionally triads: asceticism vs. mysticism, class vs. status, priests vs. prophets, traditional vs.

bureaucratic vs. charismatic authority, etc. The characteristics of ideal types emerge only in relief, in terms of the distinguishing criteria that separate one type from another. Any particular phenomenon is defined not only by what it *is* but by what it *is not*. Religion defies identification as an ideal type and consequently remains difficult if not impossible to define, precisely because the category "not religion" remains vacant. Admittedly, Asad (2003) argues that the categories of "religion" and "the secular" were historically co-constituted in terms of one another and consequently inextricably linked. However, Weber's transhistorical deployment of "religion" (but not "secularism") makes such an opposition impossible in his schema. Horii (2019) suggests that the difference is embodied in Weber's theory by the distinction between *Geist*, "spirit", and *Welt*, "world". Such a distinction indeed figures prominently in Weber's thought, but I would argue that it is deployed to distinguish between religions, in terms of the nature and extent of their engagement with or withdrawal from the world, rather than to define religion in terms of its opposite. Unlike Durkheim, who was indeed preoccupied with the universal characteristics of Religion (with a capital "R"), Weber was concerned with what distinguished one religion (or within a single religion one religious orientation) from another.

It might seem that the concept of "ideal types" would be eminently suited to the construction of typologies. On the contrary, Weber was not concerned with using ideal types for the purposes of constructing a typology of religions, an exercise that would after all have called for a definition of religion. Rather, his analysis was a compromise between history as he understood it—the explanation of unique, concrete events—and any system of abstract generalization. Weber treated specific religions, at particular moments of their development, as unique phenomena rather than as unique events. He used ideal types to construct a complex grid, where each religion occupied a place at the intersection of different dimensions. Two religions that might be similar along one dimension might be very different along others. In this way, apparent similarities were often undermined by underlying differences. For example, Weber contrasted religions that preach salvation to religions like Confucianism where notions of salvation were entirely absent. Buddhism and Christianity were both religions of salvation in this respect, but their concepts of salvation were so radically different that resemblance tended to recede into the background.

Weber was consistently concerned with demonstrating how any particular religion was different from any other, whether such religions were as far removed from one another geographically and theologically or whether on the contrary they seemed nearly—but never quite—identical. To pick but one example, in a passage that at face value seems far removed from Weber's more typical preoccupation with "world religions",[8] he elaborated the differences between Ancient Greek and Ancient Roman religions:

> " . . . the genuine Roman view concerning the general nature of the supernatural tended to retain the pattern of a national religion appropriate to a peasantry and landed gentry. On the other hand, Greek religion inclined to reflect the general structure of an interlocal regional knightly culture, such as that of the Homeric age with its heroic gods . . . While the Romans tended to regard the *ekstasis* (Latin *superstitio*) of the Greeks as a mental alienation (*abalienatio mentis*) that was socially reprehensible, the casuistry of Roman *religio* . . . appeared to the Greek as slavish fear (*deisidaimonia*)". (Weber 1978, pp. 408–9)

Typically, he situated these differences in terms of the structural features of each society as well as the particular nature of the social strata identified as the principal carriers of each religious orientation. What distinguishes Weber's approach from a sort of diluted Marxism that sees religion as an epiphenomenon of underlying socioeconomic realities is his insistence of the endurance of religious orientations over time. This is not to suggest that they remain unchanged but rather that their development is the outcome of the interplay between changing social and economic realities and the nature of underlying religiosity.

For this reason, in the play of similarities and differences that constitute the fabric of Weber's grid, the most significant resemblances are the product of genetic relations of filiation. The underlying similarities between historically related religious traditions were

never entirely overridden by the emergence of differences, however marked and however significant. For example, he employed his contrast between ancient Greek and Roman religion to explain differences between the early Church in Greek and Latin Christianity. The emphasis on individual virtuosity expressed itself as intellectualism in the early Greek Church, especially in its emphasis on theology. In contrast, the early Latin Church was characterized by relative anti-intellectualism as well as the elaborate development of a formal institutional framework. Quietly if not imperceptibly, Weber managed to weave his differentiation of ancient Greek and Roman religiosity into his narrative of the specific development of Christianity in Western Europe. On the other hand, apparent similarities between historically unrelated traditions, however striking, were ultimately far less fundamental than the differences.

In The Protestant Ethic and the Spirit of Capitalism (Weber 1992), Weber had already identified the emergence of Protestantism as a decisive moment in the formation of Western European modernity. The section on the sociology of religion in Economy and Society is a synthetic exposition of his plans to develop his project, as embodied in subsequent works on Confucianism and Taoism (Weber 1951), Hinduism and Buddhism (Weber 1958), and Ancient Judaism (Weber 1952). The overall aims of the project emerge more clearly in Economy and Society than in the individual works comprising the sections of "The economic ethics of world religions" that Weber managed to write before his untimely death. On the one hand, he was concerned with the historical processes that led to the emergence of different religions, and most particularly (but hardly exclusively) Protestantism. On the other hand, the contrast with other religions served to highlight the distinctive features of each of them, and especially of Protestantism.

For Weber, even more than for Durkheim, magic played a critical role in his historical vision.[9] Weber's vision was somewhat closer to Frazer's; magic constituted an irrational factor, and history progressed from the irrational towards the rational. However, for Weber, magic was not prior to religion but rather a constitutive feature, especially but not by any means exclusively at its origins. Moreover, his conception of rationality was vastly different. It falls entirely beyond the scope of this essay to discuss Weber's complex conception of rationality in any detail. Suffice it to say that, for Weber, rationality takes multiple and sometimes contradictory forms and that it is in no way synonymous with truth, as Frazer would comprehend it. For Weber, the history of magic is the history of "the disenchantment of the world", of the gradual and often hesitant liberation of religion from its magical components. This process of disenchantment is effected through "the systematization and rationalization of religious ethics" (Weber 1978, p. 439).[10] Weber acknowledged the enduring power of magic, embodied in his view by the Eucharist, the transubstantiation of bread and wine into the body and bread of Christ through the magic operation of priestly ritual. The possibility of absolution, even if it depended on confession, repentance, and penance, represented for Weber a barrier to the adoption of a rigorously rational religious ethic. Such rationality entailed the rigorous maintenance of ethical attitudes and behavior throughout one's entire lifetime, without the possibility of correcting occasional departures. For Weber, only two religions managed to inculcate such a fully rational ethic: Protestantism and Confucianism.

Even so, the ethical rationalities of Protestantism and Confucianism are radically antithetical. Protestantism, like almost all "world religions", is preoccupied with salvation. The notion of salvation is foreign to Confucianism, the lone exception. Historical developments leading to the emergence of Protestantism are incomprehensible without taking priests and prophets into account, even if they were ultimately abjured by Calvinists. Precisely because it is not an ethic of salvation, priests and prophets are simply irrelevant to Confucianism. The capitalist, unwavering in the active pursuit of his calling while refusing to luxuriate in the fruits of his labors, is the embodiment of the Protestant religious ethic. The cultured bureaucrat, impeccably conversant in the Confucian classics, is the Confucian counterpart. Protestantism, demanding of the faithful that they be active agents of God's will on Earth,

is intrinsically dynamic. Confucianism, requiring that its adherents correctly execute their duty so that order might be maintained, is intrinsically conservative.

The contrasts that Weber drew between Protestantism and Confucianism were symptomatic of his treatment of European as opposed to East and South Asian religious traditions. He systematically contrasted Asian religiosities to religions with their roots in the Near East: Judaism, Christianity, Islam, and to a lesser extent Zoroastrianism. The contrast was embodied in the separate development of traditions of ethical prophecy in the West and exemplary prophecy in the East. Ethical prophecy, with its origins in the unique development of post-exilic Ancient Judaism, enjoins the faithful to follow its teachings as an unconditional moral imperative. Lapses constitute sins that endanger not only the individual but also the entire community. The exemplary prophet, on the other hand, is a living demonstration of the path to salvation. It is in the believer's own interest to follow this example, but it is not in itself an ethical imperative.

The distinction between asceticism, characteristic of European religions, and the mysticism typical of Asian religions mirrored the divide between ethical and exemplary prophecy. Asceticism, Weber argued, entailed an active involvement with the world: constant labor for the glory of God combined with the rejection of earthly pleasures. Mysticism, seeking experience of the divine through contemplation, enjoined a retreat from the world. Weber was hardly unaware of mystical currents in European religion or, for that matter, asceticism in Asian religion. Reality, he argues, never corresponded precisely to ideal types. Nonetheless, the affinities between asceticism and ethical prophecy on the one hand and mysticism and exemplary prophecy on the other underpinned the contrast that Weber sought to draw between the religious foundations of Western dynamism and Eastern stasis. Of course, Weber's distinctions were far more fine-grained in his analyses of Asian as well as of European and Near Eastern religions than this global contrast between East and West. Be this as it may, the East/West divide underpinned the logic of Weber's analytical grid.

Zimmerman ([2006](#)) has suggested that Weber's comparative sociology of religion was in fact a pioneering example of "neoracism", a racism that essentializes cultural rather biological difference, in keeping with his political engagement with the nationalist *Verein für Sozialpolitik*. His work on the religions of China and India "elaborated a culturally differentiated world that did not place Europe in the position of conqueror but rather in a position of adjacent superiority" ([Zimmerman 2006](#), p. 54). Weber, however, was hardly the first to formulate such a paradigm of cultural incommensurability; credit (or blame) must be awarded to Herder, writing over a hundred years earlier ([Launay 2018](#)). Herder's work was indeed an inspiration to nationalist and often racist thought, especially in Germany, but also to anti-racist cultural relativism, for example in the anthropology of Franz Boas.[11] Indeed, Weber's personal attraction to mysticism ([Josephson-Storm 2017](#)) complicates his dichotomy between Eastern mysticism and European asceticism. Weber's dichotomizing schema, underlying his inability to define religion in unitary terms, remains fundamentally ambivalent, generating contradictory racist and anti-racist readings.

## 3. Sameness and Difference; Enlightenment and Modernity

Durkheim and Weber were both attempting to distance themselves from nineteenth-century paradigms of social evolution while not entirely freeing themselves from the evolutionary assumptions they critiqued. Theorists such as Tylor and Frazer were concerned to portray "savages" as irrational and as different as possible from modern Europeans (See [Burrow 1966](#); [Stocking 1987](#)). Durkheim's insistence that "primitive religions . . . are grounded in and express the real" ([Durkheim 1995](#), p. 2) and Weber's reluctance to adopt a teleological vision of history mark their distance from their predecessors, even as aspects of their theories nonetheless incorporated evolutionist assumptions. While Durkheim self-consciously insisted on using the term "primitive", in particular to signal his rejection of the more pejorative implications of "savagery", such language still conveyed a sense of European evolutionary superiority. Weber's paradigm of the progressive rationalization of

society, even as it acknowledged that such a rationalization might take different, indeed incompatible, forms, still embodied a teleology of sorts, albeit a less deterministic one.

Despite (or perhaps more paradoxically because of) Durkheim's passionate *laïcité* and Weber's religious unmusicality ([Horii 2019](#)), both insisted on the rationality of "religion" in general and especially on the non-European religions they used as examples, even if their choices of examples and the aim of their analyses were sharply different.[12] The question of rationality is the terrain where both thinkers demonstrate the ways in which they depart from—but equally fail to depart from—their evolutionist predecessors. Rather, Durkheim sought in Australian totemism the underlying similarities and ultimately the universal religious underpinnings of all human society. Indeed, the very notion of "civilization" is absent from Durkheim's text, though its shadowy presence is implicit in the use of the term "primitive". For his part, Weber avoided the term "primitive". Even so, the early sections of the work dealing with the origins of religion and with magic encompassed a similar domain. He even included a brief section on totemism ([Weber 1978](#), pp. 433–34). His ethnological speculations were remarkable for the absence of any empirical examples whatsoever, unlike subsequent sections that demonstrated the range of Weber's erudition. They served instead to ground Weber's analysis in terms that, as we have seen, bear superficial resemblance to earlier anthropological schemes portraying religious evolution in terms of the shift from magical irrationality to scientific rationality. Even if Weber accepted the identification of magic with irrationality, he was far more concerned with contrasting multiple trajectories of rationalization. For this reason, his examples were drawn from "civilized" societies, from the recently invented universe of "world religions" ([Masuzawa 2005](#)).

Durkheim's and Weber's theoretical analyses of the rationality of non-European religions thus revolved around the interplay of sameness and difference, in radically inverse ways. Durkheim attempted to uncover human universals—sameness—by focusing on the most-different-possible society, Australians, as a (presumably) analytically transparent lowest common denominator of humanity. Weber, on the contrary, used examples garnered from similar societies (in that all are literate "civilizations") in order to highlight the unique characteristics of modern Europe. Even so, this insistence of rationality hardly demarcates them from many of the same assumptions as their predecessors. Durkheim's proclamation of the rationality of Australian totemism does not apparently extend to the Australians themselves, as van Gennep was quick to point out. One is led to the Orwellian conclusion that all religions are rational but that some are more rational than others. The hegemony of European thought is exemplified by Durkheim's claim to be able to understand the rationality of Australian totemism in ways that were so inaccessible to Australians that they lacked the terminology to express its most basic concepts; notably, Durkheim borrows the term *mana* from other Melanesian societies. Durkheim's demonstration of putative European intellectual superiority is not quite so arrogant and dismissive as were his predecessors, but its effects are ultimately the same.

Weber does not, indeed cannot, deny agency to the people he describes, if only because the traditions he analyzes are literate. More generally, Economy and Society, as the elaboration of a sweeping theory of social action, necessarily requires conscious and deliberate social actors. Weber repeatedly refuses to attribute agency to abstractions: "class" for Marxists and, by implication, "Society" for Durkheim. Agency, of course, did not necessarily imply rationality; Weber had a great deal to say about irrational as well as rational bases for action. To the extent that Weber mirrors evolutionary schemes, it is in his vision of world history as the progressive development of rationality. To his credit, Weber does not consider this process of rationalization as exclusively European. However, he conceives of rationality as taking very different, often antithetical forms. He is not concerned with determining whether Asians are as rational as Europeans nor is it clear that such an evaluation would be meaningful or even possible. What matters is that they are *differently* rational. In religious terms, this is epitomized by the difference between exemplary and ethical prophecy. Even so, he contrasts the religious traditions and ultimately the societies of Asia as intrinsically static and those of Europe as intrinsically

dynamic, replicating assertions of European hegemony in a very different register from Durkheim.

Paradoxically, Durkheim's and Weber's contrasting frameworks for comparison, in their attempts to avoid some of the more egregious features of social evolutionary theory, instead replicate Enlightenment discourses about "savages" and "Orientals" (Launay 2018). Representations of "savages" and "Orientals" were deployed very differently, with reference to different debates about European society. "Savages" were depicted as "natural" humans, as distinct from their "corrupted" European counterparts. Rousseau among others contrasted their natural equality with egregious European inequality and injustice. Diderot emphasized the natural religion of (quite imaginary) Tahitians as opposed to the Christian denial of "natural" urges and the concomitant hypocrisy it generated. In important respects, Durkheim drew on these depictions of savagery; he was, after all, an admirer of Enlightenment thought, and Rousseau in particular, the subject of a posthumously published article (Durkheim 1918). Obviously, Durkheim adamantly rejected any depiction of "savages" as "natural"; the very point of Elementary Forms is that, like Europeans, they are eminently social beings. On the other hand, like many Enlightenment writers, he was guardedly hostile to organized religion. His political engagement as a Dreyfusard (Lukes 1973; Fournier 2013) involved an active antagonism towards a reactionary Catholicism in favor of an enthusiastic endorsement of the values of *laïcité* central to the Third Republic. Totemism, to the extent that it embodied for Durkheim a total correspondence between Society and religion, contrasted pointedly to modern European established religions that he felt no longer corresponded to social needs. Modern European religion, like much of European society, was characterized by anomie, by the lack of a firm moral compass that corresponded to the actual state of affairs. As Kuper (1988, p. 121) perceptively suggests, totemism "was the anthropologists; Garden of Eden. In contrast, the modern age was a waste land indeed".

Weber drew on a very different strand of Enlightenment thought, with its roots in Montesquieu who was singularly uninterested in "savagery" as opposed to "Oriental" systems of rule. Montesquieu elaborated a typology of systems of government that contrasted "Oriental" despotism to European systems of republican government or constitutional monarchies. For Montesquieu, Asian societies embodied a dystopic vision of egregious and arbitrary central authority, held in check in Europe only by a hereditary aristocracy jealous of its privileges. Such political comparisons are largely extraneous to Weber's sociology of religion, though they are hardly absent from other sections of Economy and Society. However, like Montesquieu, Weber never resorts to a vision of "savagery" as embodying a lost harmony with Nature or with Society but rather looks to Asia as an alternative path that, in contrast, highlights the strengths and sometimes the weaknesses of European modernity. Unlike Montesquieu, Weber does not see in Asia a vision of a possible European future to be avoided at all costs. Rather, its supposed stasis contrasts with the dynamism of Europe, embodied by the emergence of modern capitalism, a product of its religious roots in ethical prophecy and ultimately in Calvinist Protestantism. Weber's famous metaphor of the iron cage at the conclusion of The Protestant Ethic, in its disquietingly Faustian vision, prevents his contrast with Asia from being entirely self-congratulatory, but it is equally clear that, for Weber, Asia does not represent an alternative, even, as for Montesquieu, an unpalatable one.

What links Durkheim and Weber to the Enlightenment, as opposed only to their immediate nineteenth-century predecessors, is the way in which they employ representations of non-Europeans to elaborate a critical discourse on European modernity. For theorists of social evolution, their vision of non-European societies conveyed a rather smug sense of self-satisfaction and a legitimation of European hegemony. For both Durkheim and Weber, the contrasts were deployed to discern what they saw as characteristics of European modernity, characteristics that were not necessarily reassuring even as they simultaneously conveyed a sense of ultimate European superiority. In this respect, they are perfect examples of what Trouillot (1991) calls "the savage slot", the construction of representations of non-Europeans that, precisely in their contrast with European societies, serve as tropes in

discussions of Europe that have little if anything to do with the societies they purport to describe. Durkheim's and Weber's categorizations of non-European religions was, needless to say, seriously out of date, if not categorically distorted. After all, they had no first-hand experience of these religions (nor were they interested in acquiring any), relying on accounts that they (often mistakenly) considered reliable. They were not interested in non-European religions for their own sake or on their own terms. Yet, for very different theoretical reasons, they challenged paradigms that categorized such religions as intrinsically and irredeemably irrational, creating an opening if not paving the way for studying them in their own right.

**Funding:** This research received no external funding.

**Institutional Review Board Statement:** Not applicable.

**Informed Consent Statement:** Not applicable.

**Data Availability Statement:** Not applicable.

**Conflicts of Interest:** The authors declare no conflict of interest.

## Notes

1　I am well aware that terms like "savage", "primitive" and "Oriental" are now, and very rightly considered offensive. I employ them in quotation marks in the text because they are figures of the eighteenth,, nineteenth, and early twentieth century discourse.

2　Nineteenth century descriptions and analyses of Asian religions have been hotly contested more recently. However, the accuracy or even cogency of Durkheim's categorization of Buddhism does not affect the logic of his argument.

3　This explains why, to the surprise of some critics (Poggi 1972; Allen and O'Boyle 2017). Durkheim makes no attempt to trace the development of religion from "totemism" to modernity.

4　For a critical history of the construct of "primitive" in anthropology, including Durkheim's theory of totemism, see (Kuper 1988).

5　On early critical reviews of Elementary Forms, see (Jones 2013; Thomassen 2016, 2017).

6　It is precisely on these ground that Bhambra and Holmwood (2021) argue that Durkheim, unlike Weber, was not an apologist for European colonialism.

7　Weber's construct of "ideal type" is briefly discussed in Weber (1978, pp. 19–22) and at greater length in Weber (1949, pp. 89–106).

8　On the 19th century construction of "world religions" and the imperialist origins of the discipline of comparative religion, see (Masuzawa 2005; Chidester 1996, 2014).

9　The early phases of his historical narrative fall within the purview of "conjectural" history, needless to say. As outdated as they may be, they constitute an integral part of his conceptual framework.

10　However, Josephson-Storm (2017) has powerfully argued that Weber's paradigm of "rationalization" was not equivalent to his far more ambivalent attitudes towards "disenchantment".

11　Eze (1997) contrasts what he sees as Herder's anti-racism to the systematically racist thought of the Enlightenment.

12　Lukes (1973, p. 157) sketches out some of the features of this contrast.

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
