# Peer review of "Defining Religion: Durkheim and Weber Compared"

_religions, doi:10.3390/rel13020089_

Round 1
Reviewer 1 Report
This is a quite readable piece on an interesting subject. There is an admirable effort to make the piece and the discussion accessible. The author is very well informed and has a good grasp of much relevant literature, and despite dealing also with some very knotty theoretical and historical issues writes clearly and persuasively
However, the article does not really stake out an original area of research. Some of the findings and the considerations are well known and, to a great extent, obvious.
I am inclined to recommend publishing the piece. But I also think that the author
should explain better what his/her contribution to the literature is, what he/she is adding on the plan of the idea. To this end, she/he could start off with a short introduction, explaining what the article is about, what is/are the main argument of the article, what it has been already written on the topic (because otherwise the impression is that the author is the first about the topic!). To situate his/her article in a wider net, the author could see R. Forlenza, “Antonio Gramsci on Religion,” Journal of Classical Sociology 2021, which starts off with a reflection on Weber. Durkheim and Gramsci.
Other comments: the author should consider how the ethnographer Arnold van Gennep saw Durkheim’s approach to totemism and religion in his review of The Elementary Forms of the Religious Life (on this he/she could see B. Thomassen, “Durkheim’s herbarium: Situating Arnold van Gennep’s review of Émile Durkheim’s The elementary forms of the religious life,” HAU: Journal of Ethnographic Theory (2017), and “The hidden battle that shaped the history of sociology: Arnold van Gennep contra Emile Durkheim,” Journal of Classical Sociology (2016).
On Max Weber: the question of magic is crucial but there is an issue on magic/disenchantment which perhaps needs to be explored. On this, the author could see this https://maxweberstudies.org/source/files/pdfs/Szakolczai_Tribute.pdf and other works by Arpad Szacolkzai.
Finally, on Max Weber, the author should see also R Forlenza & B. Turner, “Max Weber and the Late Modernization of Catholicism,” in Edith Hanke, Lawrence Scaff, and Sam Whimster (eds.), Oxford Handbook of Max Weber (Oxford: Oxford University Press, 2020), 335-51
Author Response
I am very grateful to the reviewer for the constructive, insightful, and helpful comments. I am particularly flattered that the reviewer finds that I write “clearly and persuasively”. I have certainly tried to do so, but only readers can tell me whether I have succeeded.
I understand the thrust of the comments to be that I need to frame my argument more thoroughly, to state the overall argument clearly and formulate what I consider its contribution to the literature. To this end, I have considerably expanded both the introductory section and the conclusions. I have also doubled the number of references. Where I see my contribution – and I hope this comes out in the introduction and the conclusion – is to examine the relationship between Durkheim’s and Weber’s definitions (or non-definitions) of religion and their deployment of non-European examples. There is a considerable literature on either of these topics, but not on their relationship. The literature on Durkheim’s definition centers on the opposition of Sacred and Profane, whereas my focus is how, through his definition, he strategically sets up “totemism” as his object. Weber’s refusal to define religion has attracted little attention, and at that mostly in an attempt to discern what he took for granted rather than taking his refusal more seriously. There is also a substantial issue on Durkheim’s scholarly insufficiencies, beginning with van Gennep’s review. I am deeply indebted to the reviewer for pointing this out as well as Thomassen’s thorough contextualization. I was aware of the antagonism between Durkheim and van Gennep, but hardly in any detail, and these discussions were incredibly useful. Critiques of Weber’s approach are perhaps more recent, but no less cutting.
However, I am less concerned either with the definitions nor with the ethnographic descriptions per se than with the relationship between the two, for the ultimate purpose of constructing a critical characterization of European modernity. In this respect, they differed from most of their immediate predecessors, theorists of social evolution, whose vision of European modernity was distinctly uncritical, while in the process coming to echo very different strands of Enlightenment thought, in characterizations of “savagery” echoed by Durkheim and of the “Orient” echoed by Weber. Like most authors, I am convinced of the novelty and originality of my approach, but I must leave this to the judgment of the reviewer.
I find Szacolkzai’s distinction between magic and enchantment intriguing and particularly illuminating for Weber’s metaphor of the iron cage. However, in Economy and Society, which begins with a (for Weber) remarkably speculative discussion of the origins of religion in this-worldly magical activity, I do not find the distinction very helpful. It seems to me that Weber here really means “magic” and not “enchantment”! By and large, rationalization for Weber entails the progressive elimination of this eminently magical component.
I was asked to revise my essay during a time period when, from start to finish, my university library was totally closed due to the Winter vacation and the pandemic. As a result, and to my great regret, I have not been able to consult either of the articles by Rosario Forlenza, which I look forward to reading with great pleasure.
Reviewer 2 Report
I think the author has successfully highlighted the logical contradictions embedded in the idea of religion in Durkheim’s and Weber’s works. However, it is disappointing that the author has failed to contextualize these logical problems in relation to the colonial norms that were normalized in the milieu of Durkheim and Weber within the French Third Republic and the German Empire.
Importantly, ‘religion’ is a modern (Western and colonial) construct. It is the category that was invented and utilized by ruling elite and intellectuals in the metropoles, in order to degrade non-modern lifeways and rationalities (as ‘religious’ thus ‘irrational’ and ‘primitive’), and authorize ‘North Atlantic fictions’ of moderns (as ‘non-religious’ or ‘secular’ thus ‘rational’ and ‘civilized’) as if they represent universal order of things. Durkheim and Weber were actually part of this construction.
In this sense, it is nice to see that the author referred to Masuzawa’s work in the conclusion. However, the author does not refer to any other related academic literature which problematize the category ‘religion’ in the colonial context. Probably, a good starting point would be Nongbri’s (2013) Before Religion: A History of a Modern Concept (Yale University Press).
There is at least one existing work that examines the idea of religion of Durkheim and Weber in relation of the colonial norms of their time. I think the author should engage with the following work: Horii, M. (2019). Historicizing the category of “religion” in sociological theories: Max Weber and Emile Durkheim. Critical Research on Religion, 7(1), 24-37.
In order to understand the relationship between colonialism and social theory of Durkheim and Weber more generally, I would recommend the recent works below. These should help the author to contextualize Durkheim and Weber in a more historically nuanced way.
Farris, S. R. (2014). Max Weber's Theory of Personality: Individuation, Politics and Orientalism in the Sociology of Religion. Haymarket Books.
Bhambra, G. K., Holmwood, J. (2021). Colonialism and Modern Social Theory. Polity Press.
Kurosawa F (2013) The Durkheimian school and colonialism: Exploring the constitutive paradox. In: Steinmetz G (ed) Sociology & Empire: The Imperial Entanglement of a Discipline. London: Duke University Press, pp.188–209.
Author Response
I thank the reviewer for the thoughtful and helpful comments. I am particularly grateful for the reference to Matsushito Horii’s articles comparting Durkheim’s and Weber’s definition, which I have cited several times in my essay. While I do not concur with all of Horii’s conclusions, I have found the essay extremely helpful to me in thinking through how to develop my own argument.
I cannot agree more with the reviewer’s insistence that the article needs to be framed more fully and historically contextualized. I fully agree that “religion” is a (relatively) modern and specifically European construct, problematically and misleadingly applied to /China, India, Australia, and indeed medieval Europe. However, I disagree that it was a colonial construct “invented by ruling elite and intellectuals … in order to denigrate and degrade non-modern lifeways and rationalities.” “Paganism”, a concept with its roots in early Christian apologetics, would have been perfectly up to the task, and was deployed ferociously by Spanish missionaries in the New World. Personally, I would (and eventually intend to) argue that the concept of religion dates from the Wars of Religion in the 16th and 17th centuries and was developed in tandem with the idea of the sovereign state – but this is a debate that lies well outside the scope of my response to the reviewer’s comments. On the other hand, I very definitely concur with the reviewer that the discipline of Comparative Religion – not the same as the concept of “religion”—was an eminently colonial invention. In short, I would argue for a somewhat more nuanced approach to the colonial construction and deployment of the concept of “religion”.
In any case, my command of French social history in the late-nineteenth and early twentieth century (not to mention my ignorance of Germany) does not permit me to make any very subtle argument about historical contextualization in the metropole. (I can write with far more authority about French colonies in West Africa, but this is quite beside the point.) Where I have been able to contextualize more effectively is in the relationship of Durkheim’s and Weber’s work to that of their predecessors. I have argued that they attempted (with only very partial success) to break with the approach of late nineteenth century social evolutionists, in the process replicating different and divergent Enlightenment discourses. This is perhaps not the contextualization the reviewer had in mind.
I thank the reviewer for the references to Farris, Bhambra and Holmwood, and Kurosawa. Alas, the time I have been allotted by the Journal for revisions corresponds to a period when my university library is entirely closed for Winter vacation, not to mention COVID precautions, and I have been unable to gain access to them for the time being. I have fallen back on the resources of my personal library, and instead used Chidester, Asad, and Trouillot, hopefully to similar effect.
As the reviewer will notice, I have very considerably expanded the introductory and especially the concluding sections of the essay, hopefully answering some if not all of the preoccupations indicated.
Round 2
Reviewer 2 Report
Thank you for taking my suggestions. The article has been much improved. I recommend to accept it with minor corrections.
Overall, the author makes a persuasive argument. References to Chidester, Asad, and Trouillot are very good. However, it can be further improved by referring to some more materials. I would like to ask the editor of the journal to give the author enough time to consider the following suggestions.
I believe the core claim of the author is that Durkheim’s and Weber’s discourse on religion is part of their attempts to distance themselves from nineteenth century paradigms of social evolution. I think this claim still requires some extra supports.
In the part of Durkheim, someone like myself would still argue implicit colonial assumptions in the idea of ‘primitive’ and the religion-laique distinction in Durkheim’s writings, in which the norms of values of the French Third Republic were tacitly authorised as ‘modern’ and ‘advanced’ (as opposed to ‘primitive’) and as laique (‘public’ and ‘neutral’ as opposed ‘private’ religious faith). To counter-argue this, the chapter on Durkheim in Bhambra and Holmwood’s (2021) Colonialism and Modern Social Theory is very important. Basically, they defend Durkheim from the accusation of being a tacit imperialist.
For Weber, I would still argue that his study of religion was motivated by nationalism and his support of German imperialism. Weber believed that the success of the British Empire resulted from puritan tradition. Thus, he idealised puritan personality. I would like to know the author’s view on this.
Weber has been accused of being racist and orientalist by recent academic studies. In this light, I think the author must engage with the following works at least:
Andrew Zimmerman’s (2006) “Decolonizing Weber” (Postcolonial Studies vol.9)
Farris, S. R. (2014). Max Weber's Theory of Personality: Individuation, Politics and Orientalism in the Sociology of Religion. Haymarket Books.
Also, in my view, Weber makes sharp distinction between ‘religion’ and ‘non-religion’ in the form of the Geist-Welt distinction. He also conceptualised spirit of capitalism as a ‘secularised’ form of puritan ethic.Thus, I disagree when the author claim: “Where can one draw the line between actions that are religious or are not religious? For Weber, the question was ultimately of little or no importance.” (p. 7) I would like to know what the author thinks.
In addition, Weber was involved in magical practices in his personal life – His life and that of others around him were not disenchanted at all. Jason Josephson-Storm’s (2017) The Myth of Disenchantment discusses this matter in its chapter on Weber. It may be worth mentioning this.
I understand that the author is having a difficulty in accessing sources. However, addressing the points I made above will strengthen the author’s argument. I hope the editor of the journal allow the author to take time to reflect on these points. That would polish the article greatly.
Author Response
I thank the reader again for such helpful comments. I was able to access all the suggested books and articles with the exception of Farris’ book on Weber’s Theory of Personality. I have included references to all these works and others in the article and in the footnotes.
I have added a paragraph to address the clam that Durkheim and Weber attempted to distance themselves from the social evolutionists, Durkheim through his assertion that all religions are “true”, Weber through his avoidance if not rejection of teleological history. This said, I hope I have made it clear that I definitely would not argue that they freed themselves entirely from such assumptions, and that in important ways (Durkheim’s insistence of the “primitiveness” of Australians, Weber’s quasi-teleological insistence on the progressive development of “rationality”) they continued to reflect core evolutionary assumptions in their thinking.
Laïcité, for Durkheim, embodies the paradox of secularist religiosity. As a transcendent social value, it is squarely located within his concept of the “sacred”. Durkheim certainly promulgated the values of the Third Republic, but the alternative – reactionary French Catholicism – was just as colonialist (indeed, in Africa, even more). Communism was perhaps (and only perhaps!) the only anti-colonial alternative at the time.
The reader and I will forever disagree, I suspect, about whether Weber’s distinction (which I readily concede is critical to his thought) between Geist and Welt corresponds to the distinction religion/non-religion. I would argue that they are rather deployed as one dimension of the distinctions between different religious traditions.
With Zimmerman and others, I readily concede that Weber was an egregious nationalist. However, I would also argue that his thought is complex and contradictory, and that more generous readings are not inevitably a form of whitewashing. Josephson-Storm helps to complicate any reading of Weber. The notion of incommensurable cultural difference that Zimmerman argues is early neoracism is readily traceable back to Herder, simultaneously an ancestor of nationalist and of anti-racist thought. Contradictions abound, for Durkheim as well as for Weber. I hope I have conveyed this adequately in my paper.